# Innovation in Digital Education: Lessons Learned from the Multiple Sclerosis Management Master’s Program

**DOI:** 10.3390/brainsci11081110

**Published:** 2021-08-23

**Authors:** Isabel Voigt, Christine Stadelmann, Sven G. Meuth, Richard H. W. Funk, Franziska Ramisch, Joachim Niemeier, Tjalf Ziemssen

**Affiliations:** 1Center of Clinical Neuroscience, Department of Neurology, University Clinic Carl Gustav Carus, Dresden University of Technology, 01307 Dresden, Germany; isabel.voigt@ukdd.de; 2Institute of Neuropathology, University Medical Center Göttingen, 37075 Göttingen, Germany; cstadelmann@med.uni-goettingen.de; 3Department of Neurology, Medical Faculty, University Clinic Düsseldorf, 40225 Düsseldorf, Germany; meuth@uni-duesseldorf.de; 4Institute of Anatomy, Medical Faculty Carl Gustav Carus, Dresden International University (DIU), 01067 Dresden, Germany; richard.funk@di-uni.de (R.H.W.F.); franziska.ramisch@di-uni.de (F.R.); joachim.niemeier@di-uni.de (J.N.)

**Keywords:** multiple sclerosis, master’s program, education, multiple sclerosis management, Dresden International University, digitization

## Abstract

Since 2020, the master’s program “Multiple Sclerosis Management” has been running at Dresden International University, offering structured training to become a multiple sclerosis specialist. Due to the COVID-19 pandemic, many planned teaching formats had to be changed to online teaching. The subject of this paper was the investigation of a cloud-based digital hub and student evaluation of the program. Authors analyzed use cases of computer-supported collaborative learning and student evaluation of courses and modules using the Gioia method and descriptive statistics. The use of a cloud-based digital hub as a central data platform proved to be highly successful for learning and teaching, as well as for close interaction between lecturers and students. Students rated the courses very positively in terms of content, knowledge transfer and interaction. The implementation of the master’s program was successful despite the challenges of the COVID-19 pandemic. The resulting extensive use of digital tools demonstrates the “new normal” of future learning, with even more emphasis on successful online formats that also increase interaction between lecturers and students in particular. At the same time, there will continue to be tailored face-to-face events to specifically increase learning success.

## 1. Introduction

In neurology, there have been significant innovations in diagnosis and treatment of multiple sclerosis (MS) in the recent years [1]. Therefore, MS specialists need to be familiar with the “state of the art management” of chronic inflammatory diseases of the central nervous system. To date, however, there are no structured and industry-independent education programs for MS. Thus, a panel of MS experts and the experienced team of Dresden International University (DIU) developed the concept of the four-semester master’s program “Multiple Sclerosis Management” (MSM), which was accredited in 2019 and started in German language in 2020 [2].

This is the first time that a master’s degree program has been designed and launched around one single disease entity—a situation that does not yet exist in medical study programs or in further education studies. In addition, to date, there is no comparable study program on the market today. Either existing courses concentrate on a broader area such as neuroscience and neurodegeneration [3], immunology and inflammatory disease, neuroimmunology [4] or they address a specific target audience, such as physiotherapists [5] or MS nurses [6], or only partial aspects of MS are covered in webinars and single lectures [7]. Some specific advanced training programs are sponsored by pharmaceutical companies and are therefore not independent. The MSM master’s program offers a full and industry-independent complete package around MS, unlike scientific journals or papers for further education that usually cover only a very small aspect of pathology, symptoms or treatment and care.

In the set-up phase of the program, experts developed a variety of modules focusing on basics, clinical and diagnostic aspects, studies and statistics, therapy and rehabilitation as well as monitoring and documentation of MS. The MSM master’s program spans four semesters and is divided into six modules and a master’s thesis (Table 1).

The chronological sequence, the classification of the modules into semesters and the ECTS points to be earned in each case can be seen in Figure 1.

The module coordinators appointed for the content development of the individual modules exchanged ideas with all lecturers on the content and conceptual design of the study program several times in person and online. They selected as a team the lecturers, specified the course topics and assigned them to the teaching formats. Together with the program management of the master’s program, they also worked out the concrete time and lesson planning. In addition to the traditional knowledge transfer through lectures and tutorials by experienced MS experts, the contents of the Master’s program are to be taught with a particularly high practical component. For this purpose, preceptorships in specially selected MS centers, excursions and regular journal clubs as well as digital case conferences serve the direct practical implementation of the learned contents on site.

However, the start of the Master’s program coincided with the beginning of the COVID-19 pandemic, so that it had to be conducted online to an even greater extent than planned. The existing plans could not be applied and new concepts had to be designed and implemented at short notice in an “emergency mode”. Due to the COVID-19 pandemic situation, the problem arose to shift all modules where it was possible from face-to-face learning to digital format. Although in the meantime many studies exist comparing digitally presented lectures and courses vs. face-to-face learning [8,9,10,11,12,13], DIU has been intensely concerned with the acceptability of a “digital only” education.

What remains of the creative digitization push, born out of necessity that has changed the image of universities so much? This paper provides an introduction and examines the innovative use of a cloud based digital hub (Microsoft Teams) for computer-supported collaborative learning in the MSM master’s program as well as student evaluation of the program. In addition, the authors consider the extent to which the predominantly online master’s program can successfully teach the complex, dynamically changing scientific work content in a way that is adapted to different levels of knowledge. Specifically, the authors take a look at the use cases of computer-assisted collaborative learning, the technical support provided by the organizers and the course instructor, the quality of the master’s program content transferred to the virtual version, the performance of the instructors, and the students’ interactions with the instructors are considered.

## 2. Materials and Methods

### 2.1. Use Case Analysis of Computer-Supported Collaborative Learning

Microsoft Teams [14,15] is used as a technological basis for the MSM Master’s program in learning, teaching, collaboration, and cooperation processes, which has proven particularly effective during the COVID-19 pandemic. Since the services are cloud-based and software updates are provided automatically in the so-called “evergreen mode“, there is no need for technicians and IT teams to support the platform itself after the initial setup. Microsoft Teams serves as a digital hub that brings together conversations, content, tasks and apps in one place. The digital hub provides extensive security and compliance-specific features that will not be discussed here. Rather, the focus is on the analysis of innovative use cases that have been implemented within the MSM Master’s program: one central data platform for highly effective organization of the Master’s program, online classrooms in a distance learning environment for synchronous and self-directed asynchronous learning, flexible knowledge transfer in the learning video portal, and establishment of special learning areas for peer-to-peer learning. The use cases present possible applications of certain tools for students and lecturers in the pandemic situation and show how they were implemented.

### 2.2. Systematic Evaluation of Teaching and Student’s Feedback

#### 2.2.1. Participants

DIU program managers conducted the evaluation on a qualitative and quantitative level and asked the participants of the MSM master’s program to share their experiences with module 1 (“Theoretical Principles”, details see Table 1) in the form of qualitative feedback and to complete a standardized evaluation questionnaire (quantitative feedback) after module 2 (“Clinical & Diagnostic Aspects”, details see Table 2). Participation was voluntary in both cases.

#### 2.2.2. Data Collection

For module 1, students were able to report their experiences via email or Microsoft Teams. As a result, the feedback providers were known, allowing specific queries for the further development of the program. For the evaluation, the responses were then aggregated and anonymized. In the further course, DIU program managers asked students to complete a systematic and anonymous evaluation at the course and module levels. For this purpose, DIU program managers used in-house, already established standardized questionnaires that are used for evaluation in all study programs at DIU [16]. For module 2, students answered an evaluation questionnaire with 23 questions in 5 categories (Table 2), which they could answer with values on a scale of 1 (strongly agree) to 6 (strongly disagree). In addition, students had the opportunity to indicate in free text fields what they particularly liked or disliked about the course, what they would recommend to improve the quality of the course and the lecturer’s performance.

#### 2.2.3. (Statistical) Analyses

Authors used the Gioia method [17] to evaluate the information from the feedbacks for module 1, some of which were very detailed. The Gioia method allows for qualitative evaluation with inductive and summary category building, allowing for creative influence with systematic accuracy. It assumes that the organizational world is socially constructed and its participants are knowledgeable individuals who can explain their intentions, thoughts and actions.

For module 2, authors calculated means and standard deviations to describe the student population and evaluation variables and used charts for illustration. In addition, they screened the free texts for concise statements, which are presented as examples in the results.

## 3. Results

### 3.1. Use Cases for Computer-Supported Collaborative Learning

The COVID-19 pandemic has quickly changed professional and private life, and a tremendous need emerged to hold meetings exclusively online, organize video conferences or create videos for lessons and further education. The lecture halls, meeting areas and learning spaces were empty (Figure 2). Teaching and learning shifted to virtual space under high time pressure, leading to a variety of innovative use cases, that were described and analyzed using the MSM Master’s program.

#### 3.1.1. Single Point of Truth for a Highly Effective Organization of the Master Program

All relevant information for the organizational management of the study program is available on one single data platform (example view in Figure 3). This includes, for example, all study documents, timetables, applications, forms, support information, step-by-step instructions, relevant literature, etc. The exchange with the program management and the lecturers is chat-based and transparent for all members. This dramatically reduces the effort required for bilateral communication and email.

#### 3.1.2. Online Classroom in a Remote Learning Environment for Synchronous and Self-Directed Asynchronous Learning

Students have access to a wide range of tools and resources for remote learning via the digital hub (example view in Figure 4). Lecturers present their content as live lectures and can use functionalities for synchronous learning such as file sharing, various forms of participant feedback and real-time interaction or group workspaces. In addition, all documents for a course are available in chronological order. This gives students the flexibility to study the content at their own pace, at their own time and with their own device (asynchronous learning).

#### 3.1.3. Flexible Knowledge Transfer in the Learning Video Portal

The digital hub allows synchronous lessons, lectures and events to be held in special channels and to securely share and interact on video content from presentations. Microsoft Stream, the video service from Microsoft Teams [18], makes it possible to create live lectures, record them automatically and make them available regardless of location and time. The app simplifies uploading, organizing, and sharing video content across the Master’s program. Students call up the recording of a missed learning session or recall session at a time of their choice in the video portal (Figure 5).

#### 3.1.4. Set up of Special Learning Areas for Peer-to-Peer Learning

Joint activity learning areas are presented in an organized, structured, and consistent manner. In the MSM master’s program, there are two light house examples of how specific learning areas can be established: case conferences and journal clubs. In the case conference students present their own case or—if they do not have patient contact—a published case to the lecturers and the other students (peer-to-peer-learning). In journal clubs, they discuss professional articles and several students can participate in analyzing a single article. In this process, participants discuss all articles according to fixed criteria: background, method, results, discussion and conclusion. The case conferences and journal clubs are organized using the wiki functionality in Microsoft Teams (Figure 6). Students autonomously enter their contributions into the given schedule grid and provide the information to be presented online. These approaches also allow lecturers to assess students’ learning and experience more deeply regarding their areas of interest.

Since the start of the course, the platform has been used regularly by the students and the amount of learning content is continuously growing. In the future, exams will also take place on the digital hub and the lecturers will use the digital hub to supervise the preparation of master’s theses.

### 3.2. Evaluation of Student’s Feedback

#### 3.2.1. Participants

Most of the 19 participants (89%) in the first matriculation of the Master’s program are physicians with advanced training in neurology, but there are also biologists. Slightly more than half (53%) of the 19 students are women, which means that there is a balanced gender ratio. Students are on average 39.4 ± 8.9 years old, ranging from 28 to 60 years.

Four students gave very detailed feedback on module 1. For module 2, the participants of the study program evaluated a total of 36 courses with regard to content, structure, and set-up of the course, with regard to the lecturer, methodological aspects, and quality and use of course materials. Response rates ranged from 16% to 84%, meaning that not all course participants always rated each single course. Since the master’s program is currently still running and not all modules have been completed or started, not all modules could be evaluated yet.

#### 3.2.2. Evaluation Survey

Four students gave detailed feedback on module 1 with 55 statements. There were 24 statements on the aggregated dimensions “Communicative and didactic quality of teaching” and 15 statements on the topic “Difficulties in studying”. For example, students praised the “communicative and didactic quality of teaching”: “In my opinion, everyone was a real asset in their own way and I found the often very different presentations and lectures very good throughout.”, and the interactivity: “The highlight of module 1 for me was the opportunity for interaction”. “Difficulties in studying” were addressed by two students and included, in particular, the large amount of time needed to rework the learning materials for certain groups of participants with non-medical backgrounds: “Especially for me as a non-neurologist, the module was also a good introduction; but also demanding and a lot of reworks was needed.” Other statement dimensions related to the quality of teaching and the provision as well as the practical or research relevance of the content, the commitment of the university and the lecturers to the students, the organization of teaching and the examination system were rated positively overall.

In module 2, students evaluated 36 lectures (status: May 2021). They rated all evaluation categories with a mean of 1.2 or 1.3, indicating strong or certain agreement with the respective items, which the authors interpret as high satisfaction with the respective topic. Means and standard deviations for evaluation categories are presented in Table 3.

In the category “content, structure and organization of the event”, the students gave strong to certain agreement that the objectives of the event were clearly recognizable (1.26 ± 0.74) and that the content structure (“red thread”) of the overall event was sensible (1.32 ± 0.77), as well as that the event time was used in a way that promoted learning (1.37 ± 0.81). For the students, the relevance of the content covered for practice became clear (1.31 ± 0.75) and they were able to contribute their personal competencies and previous experience appropriately (1.47 ± 1.01). In category “lecturer” the students gave strong to certain agreement that the lecturer has stimulated the discussion of the topics (1.27 ± 0.74), emphasized active participation of the students (1.37 ± 0.89), succeeded in making the event appealing (1.31 ± 0.76) and was appreciative in dealing with students (1.15 ± 0.56). The students also rated the “methodical aspects” very highly. They gave strong to certain agreement, that teaching/learning forms (individual, partner, group work, work in plenary) were appropriate (1.42 ± 0.86), that the lecturer was able to present complex content in an understandable way (1.27 ± 0.72) and gave appropriate feedback or responded appropriately to the group (1.32 ± 0.79). The quality of the media content (presentations, scripts, exercise sheets, e-lectures, etc.) was appropriate (1.32 ± 0.71) and the media and (online) tools used were used sensibly (1.27 ± 0.69)—students gave strong to certain agreement to these items in category “documents, course materials and media: design and use”. Finally, students rated positively the aspects of “technical support of the online events”—they gave strong to certain agreement with the items satisfaction with technical support (1.09 ± 0.31) and supervision during the courses (1.21 ± 0.43), and the suitability of the virtual classroom for the course (1.26 ± 0.54).

Figure 7 shows the proportions of agreement in a stacked bar graph, clearly showing the large proportions of strong and certain agreement with the items. A percentage of 75% to 85% of students strongly agreed with each item, indicating a high level of student satisfaction with the implementation of module 2.

In the free text ratings, the students gave a lot of praise regarding the content and the competence of the lecturers, e.g., “extremely exciting topic”, “broad coverage of the subject”, “very interesting, practice-oriented presentation of the clinical pictures” or “Prof. XY gave a very clear and comprehensible lecture with many good examples” and “Prof. XY managed to give an exciting and very informative lecture and at the same time to emphasize the relevance of this topic, which is rather neglected in the neurological study of MS”. However, the students also criticized the speed of presentation and comprehensibility: “the topics of motor disorders and pain in MS came far too short”, “very fast pace in the presentation of some studies”, “unfortunately, the lecture was very technical and not very didactically prepared” or “The breakdown of the technical approach to the students’ world of understanding is only partially successful”. The students also made suggestions for improving the quality of the course as well as the performance of the lecturer, exemplified by the following: “the questions asked in between were good, could be made interactive and use the ‘mentimeter’ [app for real-time feedback] for example”, “The material should be distributed over two lectures or another lecture [...] should be planned”, and “It would be nice for future years of study (not possible this time due to Corona) to hold tutorials in a classroom context in order to further promote active engagement with the topic and especially the exchange in the group”.

## 4. Discussion

This paper explored the innovative use of a cloud-based digital hub for computer-supported collaborative learning in the MSM master’s program as well as student evaluation of the first semester of the master’s program, considering the challenges of the COVID-19 pandemic.

Since the MSM master’s program is aimed at professionals in neurology, it was planned from the beginning with a strong online component. Due to the start of the master’s program in the midst of the COVID-19 pandemic, the organizers had to very quickly adjust the teaching formats towards even more online courses. In a short time, DIU succeeded in establishing Microsoft Teams as a cloud-based digital hub and technical basis as well as a central teaching, learning, communication and cooperation platform, which proved to be very effective. The centralized data platform served the highly efficient organization of the master’s program. Thus, online classrooms were available in a distance learning environment for synchronous and self-paced asynchronous learning. The establishment of a learning video portal and special learning areas for peer-to-peer learning made flexible knowledge transfer possible. However, this master’s program benefited from digitization not only in learning and teaching, but also through the opportunities for close coordination between the lecturers and the course management with the academic management and module coordinators, as well as among the students themselves.

For the first two modules of the program, the authors collected student feedback and analyzed it both qualitatively and quantitatively. The students rated the courses in the modules and the modules as a whole as good to very good. They were very satisfied with the content of the courses, with the knowledge transfer by the lecturers and with the interaction with each other, as well as with the lecturers. Some of the students wished for more time for certain topics, more interaction with lecturers and would have liked to have covered some specific topics such as magnetic resonance imaging (MRI) on-site in a face-to-face event. Such face-to-face events were also planned, but due to the circumstances of the COVID-19 pandemic, they could only be held as online events.

Nevertheless, authors indicate some limitations in the interpretation of the results. For example, there is a relatively small underlying response rate for individual courses, which is unfavorable for evaluation given the already small number of participants in the program. However, it is important to keep in mind that the evaluation of the courses and modules also took place under pandemic conditions, and students’ ambitions to evaluate the courses online may not have been as high after a day full of online events. It should also be noted that the evaluation of the first two modules is only part of the evaluation of the program. The evaluation of the other modules and the entire master’s program by the students is still pending. In addition, there is no comparative data on student satisfaction with the quality of the master’s program under “normal conditions” because it was not implemented in the period before the COVID-19 pandemic. Therefore, whether this master’s program will improve MS therapy and make a valuable contribution to the scientific advancement of the entire MS field remains to be seen.

The present work shows that, despite the aforementioned challenges, the MSM master’s program is proving to be a great success, not least because of the fruitful interactions between lecturers and students. In addition, there are a few learnings that will promptly inform further implementation of the program.

Only the widespread use of digitization and digital tools made it possible to respond quickly to the imposed changes in the face of the COVID-19 pandemic and to effectively implement adjustments and necessary rescheduling. Above all, however, the “emergency mode” provided many insights and hints into the future “new normal”. The pandemic showed the limitations of a traditional “bricks and mortar” university and highlighted the growing importance of using online tools. At the same time the value of a physical place for learning and teaching became very clear. On-site learning in presence will have in future a very special quality and will be of particularly high value. Certainly, the planned on-site portions of the program in specialized clinical settings or active participation in expert meetings will help open new perspectives for students [19]. By using digital tools a new format has gained more attention: the “flipped classroom”, where the students actively work on the contents during the knowledge transfer stage before interacting with the lecturers and peers where they assimilate what they have read, watched or otherwise attempted [20,21,22].

The use cases for computer-based collaborative learning implemented in the first two modules of the master’s program will be expanded and applied in the remaining modules. The digital implementation of the MSM master’s program enables a “learning study program” using a rapid implementation of the PDCA (Plan–Do–Check–Act) cycle. Agility as one of the important current megatrends thus offers a high practical value for the MSM master’s program.

As a further learning, lecturers are encouraged to use less of a pure presentation style to deliver content, and to interact and share even more with students at eye level. New competencies for the lecturers in developing attractive didactic formats are required as well as a new understanding of the role of module coordinators in the digital world are necessary: An important lesson learned is that when the MSM master’s program is carried out digitally, there is a special requirement for the module to be accompanied by a person as a learning coordinator responsible for the module. As learning coordinators, they are tasked with guiding adults in their professional development.

## 5. Conclusions

Although the MSM master’s program was launched under pandemic conditions and the associated challenges, it was and still is possible to design, implement and continuously adapt a completely new, disease-centered study program thanks to a flexible online platform. Student response and feedback to date demonstrate both the high quality of the program and the potential of the Master’s program to make an important contribution to the MS field.

Based on this extremely positive experience, an internationalization of the program is planned to allow neurologists and other interested parties from other countries to access this high-quality Master’s program. The program will then be offered in English. Another idea for the future is to build a video platform for the public for knowledge around the MS. In addition, new opportunities for content, technical and didactic improvement as well as new digital developments are to be constantly used to further develop the Master’s program. This also includes the technology of the digital twin, which will find its way more and more into patient care in the coming years. For MS patients, the digital twin is an important step towards innovative and individual disease management. A digital twin for MS is a digital image of the MS patient—paired with the patient’s characteristics, it allows health care professionals to process large amounts of patient data. This can contribute to more personalized and effective care by integrating data from different sources in a standardized way, implementing individualized clinical pathways, supporting doctor–patient communication and facilitating shared decision making [23].

## Figures and Tables

**Figure 1 brainsci-11-01110-f001:**
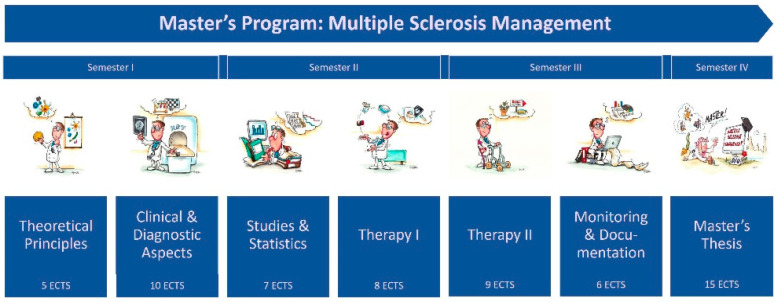
Timetable for modules in MSM master’s program with indication of ECTS credits. The MSM cartoons were created by Phil Hubbe.

**Figure 2 brainsci-11-01110-f002:**
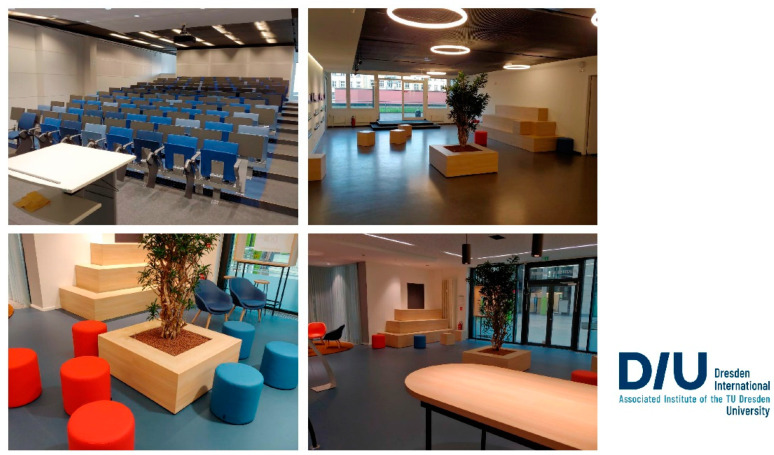
On-site teaching and learning spaces at Dresden International University.

**Figure 3 brainsci-11-01110-f003:**
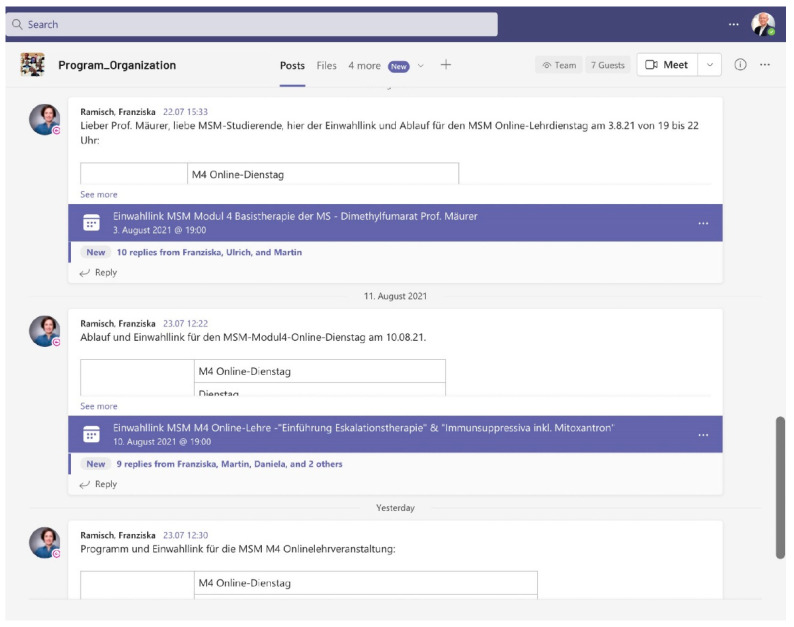
Communication of access information to online events as single point of truth.

**Figure 4 brainsci-11-01110-f004:**
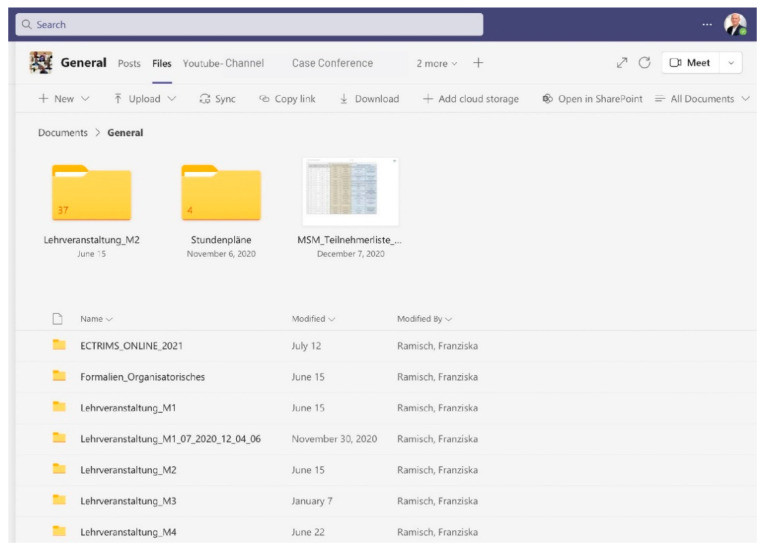
Learning modules that fit into vocational qualification.

**Figure 5 brainsci-11-01110-f005:**
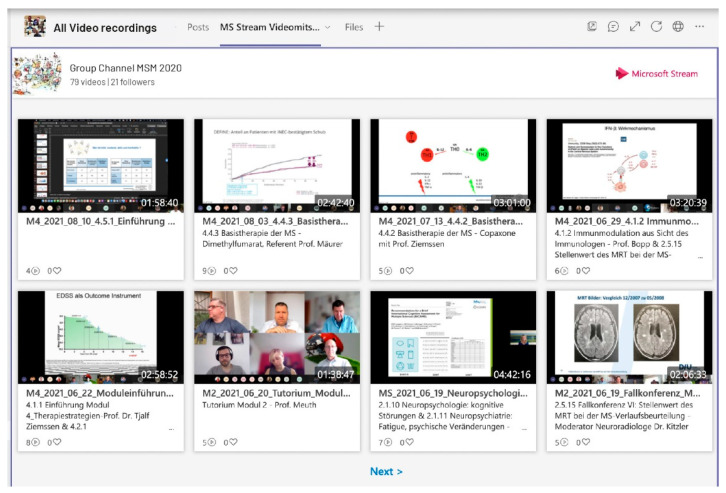
Live streaming and video on demand portal.

**Figure 6 brainsci-11-01110-f006:**
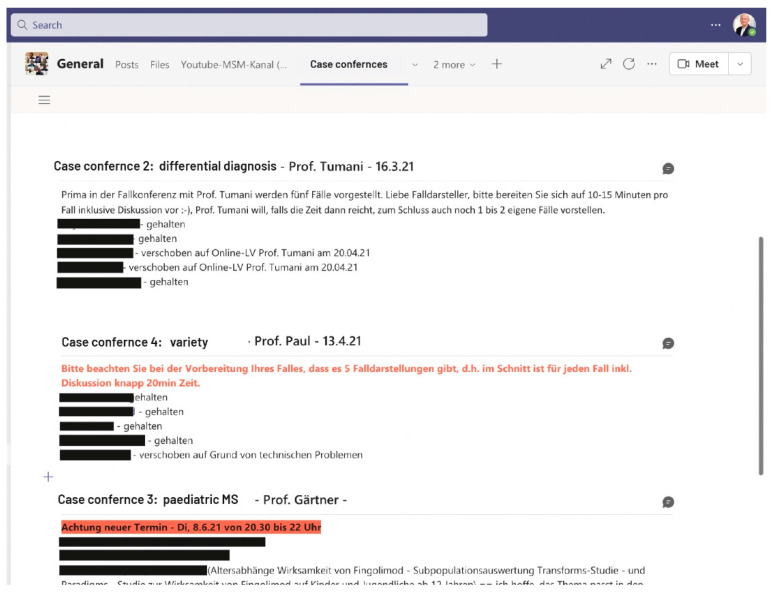
Students work and learn on a project basis.

**Figure 7 brainsci-11-01110-f007:**
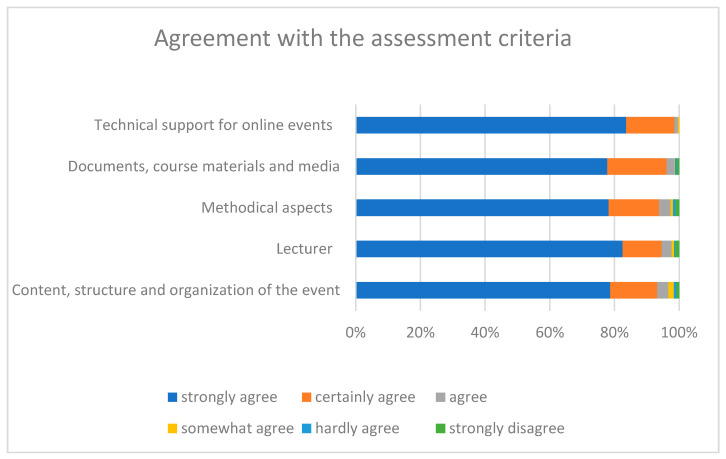
Proportions of satisfaction with the implementation of module 2.

**Table 1 brainsci-11-01110-t001:** Modules and topics in MSM master’s program.

Modules		Topics
1	Theoretical Principles	- basics and epidemiology of MS - factors of diagnosis and therapy - immunological basics - basics of pathology and pathophysiology - therapeutic interventions - methods for disease monitoring
2	Clinical & Diagnostic Aspects	- differential diagnostics - cerebrospinal fluid and blood tests - image diagnostic procedures - functional effects of demyelination - neurophysiological, neuropsychological and neuro-urological examination procedures
3	Studies & Statistics	- evaluation and application of study designs - selection of statistical tests - interpretation of results - analysis of real-world data in the context of MS practice
4	Therapy I	- differences between the therapy of acute relapses and a disease-modifying or progression-modifying therapy of MS - weighing of indications and patient profiles - successful therapy strategies for individual patients
5	Therapy II	- non-drug procedures to treat disease-associated symptoms - goals and implementation of symptomatic and complemetary therapies - neurocognitive and psychological interventions - rehabilitative and palliative medical measures
6	Monitoring & Documentation	- patient documentation - individual monitoring according to standards and therapy goals - health economic aspects - new possibilities in the field of e-health - MS-specific networks - associations and registers - big data
	Master’s Thesis	- Master’s thesis or scientific paper in a peer-reviewed or PubMed-listed journal (thematic review, meta-analysis, original scientific paper) - topic is submitted by the student and finalized by the scientific director of the program

**Table 2 brainsci-11-01110-t002:** Categories of evaluation questionnaire and the corresponding items.

Category	Questions
Content, structure and organization of the event	- The goals of the event were clearly identifiable. - The content structure (“red thread”) of the overall event was sensible. - The relevance of the contents covered for practice became clear. - Course time was used in a way that was conducive to learning. - Students were able to appropriately contribute their personal competencies and prior experience.
Lecturer	- The lecturer has stimulated the discussion of the topics. - The lecturer emphasized active participation of the students. - The lecturer is appreciative in dealing with students. - The lecturer succeeded in making the event appealing.
Methodical aspects	- Methods and teaching/learning forms (individual, partner, group work, work in plenary) were appropriate. - The lecturer was able to present complex content in an understandable way. - The lecturer gave appropriate feedback or responded appropriately to the group.
Documents, course materials and media: design and use	- The quality of the media content (presentations, scripts, exercise sheets, e-lectures, etc.) was appropriate. - The media and (online) tools used were used sensibly.
Technical support for online events	- I was satisfied with the supervision and support during the digital course. - I was satisfied with the technical support. - The virtual classroom was suitable for the course.

**Table 3 brainsci-11-01110-t003:** Average score of satisfaction.

Category	Mean ± Standard Deviation
Content, structure and organization of the event	1.35 ± 0.80
Lecturer	1.28 ± 0.75
Methodical aspects	1.33 ± 0.79
Documents, course materials and media: design and use	1.29 ± 0.70
Technical support for online events	1.18 ± 0.44

## Data Availability

The data presented in this study are available on reasonable request from the corresponding author.

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
