# Peer review of "Innovation in Digital Education: Lessons Learned from the Multiple Sclerosis Management Master’s Program"

_brainsci, 2021, doi:10.3390/brainsci11081110_

Round 1

Reviewer 1 Report

This is an interesting paper regarding the urgent fully digital transformation of a master program due to the current pandemic . Authors have used primary results of surveys conducted to the students regarding their satisfaction of this program. First of all, it seems that this is a master program of a very high quality. The authors have succeded in fully digitalising this master program instead of just doing digital platform-based lectures. Regarding the manuscript, i do have some minor comments.

  1. I would suggest the authors to review the manuscript regarding MDPI style format.  i think that e-mails of the authors should be presented in the affiliation sections. In addition , i do not think that in the abstract the numbers into brackets and the names of subsections such as methods , results, etc. should be shown since this should be an unstructured abstract.
  2. Microsoft team and stream should be referenced.
  3. The authors state that they plan to do this master internationally. Do they plan to use other language rather than German?
  4. How the authors ensured the participants that their answers were anonymus? Generally, one would expect that since personnal user names and passwords would be used in order to enter the platform, the anonymity of their answers might be compromised and this might provide bias to the results. Please further explain the procedure in order to rate these modules and the procedure of collecting these data anonymously.
  5. There are some figures and screenshots that in my opinion could be ommited. The first figure presenting the lecture halls of the university seems to be unecessary since this paper describes a fully digital program. In addition screenshots used for figure 2, 3, 4, 5 could be improved or ommited, since they do not add any value to this paper.
  6. The presentation of the program ( table 2 and figure 6 ) could be presented earlier than the methodology of the questionnaires, perhaps in the introduction section.
  7. Since this program has not ran in the pre COVID era, there is no data to compare regarding the students satisfaction of the quality of this master. This could be added as a limitation. 
  8. The authors could discuss that by digital transforming such programs, there is also need for technicians and IT teams , that should closely work with the academic team in order to provide platforms such as this one described.

Overall, this is an interesting paper with preliminary results that succesfully addresses the current challenges in academic teaching during this pandemic.

Reviewer 2 Report

This is article provides valuable insights to the digital transformation of an postgraduate course in multiple sclerosis. The manuscript has a nice flow and is comprehensive. I have a couple of minor but compuslory issues:

  • Lane 131: „massive hype“ needs to be rephrased to „needs“
  • Lane 133: „like empty“ implies something different from what the authors want to say
  • Lane 138: clarify the meaning of „innovative use cases“
  • Copy paste figures from a German homepage do not make sense if the language of the manuscript is in English
  • There is a mixup of tenses, jumping from active to passive voice. Harmonize the entire manuscript with the help of a native speaker
